# Coverage of climate change in introductory biology textbooks, 1970–2019

**Rabiya Arif Ansari**[1]◐, **Jennifer M. Landin**[ID][2]◐*

**1** College of Agriculture and Life Sciences, North Carolina State University, Raleigh, North Carolina, United States of America, **2** Department of Biological Sciences, North Carolina State University, Raleigh, North Carolina, United States of America

◐ These authors contributed equally to this work.
* jmlandin@ncsu.edu

**Data Availability Statement:** All data files are available from the DRYAD database (https://doi.org/10.5061/dryad.tht76hf2f).

**Funding:** The author(s) received no specific funding for this work.

## Abstract

Climate change is a potent threat to human society, biodiversity, and ecosystem stability. Yet a 2021 Gallup poll found that only 43% of Americans see climate change as a serious threat over their lifetimes. In this study, we analyze college biology textbook coverage of climate change from 1970 to 2019. We focus on four aspects for content analysis: 1) the amount of coverage, determined by counting the number of sentences within the climate change passage, 2) the start location of the passage in the book, 3) the categorization of sentences as addressing a description of the greenhouse effect, impacts of global warming, or actions to ameliorate climate change, and 4) the presentation of data in figures. We analyzed 57 textbooks. Our findings show that coverage of climate change has continually increased, although the greatest increase occurred during the 1990s despite the growing threats of climate change. The position of the climate change passage moved further back in the book, from the last 15% to the last 2.5% of pages. Over time, coverage shifted from a description of the greenhouse effect to focus mostly on effects of climate change; the most addressed impact was shifting ecosystems. Sentences dedicated to actionable solutions to climate change peaked in the 1990s at over 15% of the passage, then decreased in recent decades to 3%. Data figures present only global temperatures and $CO_2$ levels prior to the year 2000, then include photographic evidence and changes to species distributions after 2000. We hope this study will alert curriculum designers and instructors to consider implicit messages communicated in climate change lessons.

## Introduction

Climate change greatly affects human society and the earth's ecosystems. Given the outsized contribution of greenhouse gasses from developed nations, and specifically the United States, analyzing climate change curricula can inform educational policy and practice. In this paper, we use document analysis of U.S. biology textbooks to explore the content, organization, and innovations of climate change coverage over 50 years.

**Competing interests:** The authors have declared that no competing interests exist.

## Climate change impacts on society and the environment

Climate change impacts human health, agriculture, weather patterns, infrastructure, and extinction rates. The World Health Organization states, "Climate change is expected to cause approximately 250,000 additional deaths per year, from malnutrition, malaria, diarrhea and heat stress" [1]. Since the 1980s, each decade has been warmer than the previous one, with 2021 recorded as the sixth hottest year since planetary temperatures have been recorded [2]. Extreme temperatures account for more than five million deaths each year globally, making up 9.4% of all deaths between 2000 and 2019 [3].

According to the UN Office for Disaster Risk Reduction, economic losses have increased by 151% due to climate-related disasters. The total economic loss from extreme heat in the United States is over $100 billion annually due to productivity losses; this number is predicted to rise to $500 billion annually by 2050 [4].

Sea level has risen 21–24 cm since 1880 [5]. Kulp and Strauss estimate that 630 million people live on land projected to sit below annual flood levels by 2100 [6]. Flooding due to climate change also impacts food and water supplies as saltwater intrusion endangers access to freshwater. The global urban population facing water scarcity is projected to increase from 933 million people in 2016 to 2.4 billion people in 2050 [7]. Climate change is expected to reduce maize crop yields by 24% as early as 2030 [8]. Román-Palacios and Wiens warn that if climate change continues to increase at the current rate, up to 30% of their study's 538 animal and plant species may face extinction within 50 years, even with an expected shift in the species' niches [9].

Americans are among the top emitters of carbon dioxide per capita. Because of the severe climatological, ecological, and societal effects of greenhouse emissions, understanding the process of climate change and options to limit or resolve it should be a priority for the United States.

## Climate change in educational settings

Educational topics, and the time devoted to them, often change depending on the knowledge and needs of society. For example, Gangwani and Landin found that insect-based lessons declined in U.S. textbooks after the 1950s, when pesticide use eased the agricultural and health effects of insects on American society [10]. Textbooks shifted to expand molecular biology topics after 1960 following seminal DNA and genetics research publications [11]. Textbook figures of cell anatomy changed following the publication of a revised illustration in *Scientific American* in 1961 [10]. Given the serious effects of climate change, its growing influence over human society, the high impact of U.S. carbon emissions, and the massive increase in research and publications related to the topic, we would expect comparable coverage of the topic in American educational settings.

Science teachers exhibit denial of, or confusion about, climate change. Plutzer et al. extensively studied teachers' classroom methods and educational backgrounds pertaining to climate change [12]. They found that, while most teachers cover the topic, 31% report sending explicitly contradictory messages in effort of teaching "both sides." Almost one-third of the teachers emphasized that recent global warming is "likely due to natural causes," and 12% do not emphasize human causes at all. Fewer than half of teachers reported formal education in climate change in college.

Only 29 U.S. states and the District of Columbia have educational standards that address anthropogenic climate change. The National Council for Science and the Environment evaluated state science standards in 2020; while Wyoming was the only state to receive an A for how it addressed climate change, 10 states received a D or worse, including populous states such as Florida and Texas [13].

## Science textbooks as historical documents

Textbooks are important documents to study which information society deems valuable and appropriate [14]. Textbooks represent an authoritative source in education; most teachers identify textbooks as the largest influence on content selection, especially in the sciences [15]. Therefore, analyzing texts for topic coverage and organization, language use, and pedagogical recommendations can inform us of the intentions of curriculum experts, as well as the influences of society or culture.

The sequence of chapters can play a critical role in how, or if, content is addressed in the classroom. Controversial topics of reproduction, evolution, and conservation were placed at the rear of biology textbooks in the 1930s, beginning a tradition of book organization still observed today [16]. Instructors usually progress through a textbook from one chapter to the next and, in studies examining teachers' use of textbooks, chapters at the end of the books are frequently skipped [17, 18].

Visual elements, such as graphs, maps, and photographs, play a positive role in students' processing, organizing, and recalling textual information [19]. Complex data visuals, incorporated into the text, improve students' problem-solving and memory of facts [20]. Therefore, the data visuals displayed within the climate change passage should indicate content valued for this reinforcement.

## Predictions for textbook analysis

If curricular decisions are based on progress toward an ideal goal, then we should see movement toward this goal over generations of educational documents. In this case, the goal is a conceptual understanding of the greenhouse effect, the impacts of increasing greenhouse gasses, and immediate behavioral, political, and cultural changes to slow or stop global climate change.

Our content analysis research focuses on any differences between this ideal goal and the operational curriculum used by teachers and students. While there are multiple other factors influencing the perception of climate change, textbooks are valued for the authoritative roles they play in education systems; this would make the textbook a persuasive instrument in the fight against climate change.

We reviewed climate change passages in textbooks published between 1970 and 2019 and predicted that:

1. The amount of climate change coverage will increase.
   Since textbooks reflect societal values, educational goals, and expert knowledge, we predict that the length of the climate change passage in introductory biology textbooks will increase substantially over time. Scientists since the 1970s have gained more understanding of the topic of climate change, and the public's awareness of the issue has grown as environmental conditions have worsened and the urgency of prevention or restoration increased.

2. The passage addressing climate change will move forward in the textbook.
   We expect that the sequence of chapters will transition from the traditional end-of-book placement of conservation issues to shift earlier in the book, indicating increased importance, decreased perception of controversy, and a reduction in the likelihood that instructors will omit the content.

3. The climate change passage will shift from a description of the greenhouse effect to incorporate coverage of effects and solutions over time. The solutions will also shift from a national or international focus to include productive individual or local actions.

These shifts reflect the increased rate of change in temperatures and weather patterns, and severity of risk. Organizations focused on solutions have more recently emphasized individual, or grassroots, actions as drivers for political or industrial change. Examples include Article 12 of the Paris Agreement (2015) and founding of grassroots climate change organizations such as 350.org (2008), Project Drawdown (2014), and School Strike for Climate (2018).

4. The number of data figures referenced in the climate change passage will increase.
An increase in the number of visual elements, especially those portraying data, corroborates statements presented in the text, promotes understanding and recall of climate change evidence, and increases the space devoted to the topic of climate change. Therefore, the number of figures should reflect the value placed on the content.

## Materials and methods

We used an intentional sample of the most widely used textbooks in U.S. undergraduate education. Gangwani and Landin determined that high-edition biology textbooks were representative of a larger sample of textbooks, so our sample primarily included books with three or more editions [10].

### Textbook selection

We located 57 college-level introductory biology textbooks published between 1970 and 2019. We selected 1970 as the starting publication date to ensure detection of the earliest coverage, as a scientific consensus on global warming was reached in the 1970s [21]. Books written for different college audiences (science and non-science majors) were combined in our sample since this separation is a recent phenomenon. Analyzed textbooks had three or more editions, except for two books published in the 1970s, as fewer textbooks reached high edition numbers during that decade. Giordano addresses this phenomenon as a reflection of societal factors, such as a substantial increase in the cost of textbooks, mergers among publishers, a backlash against textbooks in general, and removal of textbooks due to racial and gender bias or the perceived controversy of inclusivity [22]. Due to little coverage of climate change in this decade, we included low-edition textbooks to ensure a larger sample.

To confirm that our sample of textbooks included those most often used in higher education, we contacted library reference desks at the three largest universities in the states of California, North Carolina, and Texas to request a list of textbooks used in their introductory biology courses. While the lists were limited to the past decade, all books had been included in our sample (see S1 Appendix).

### Passage selection

The passage addressing climate change was located by searching for key phrases in the index: atmospheric carbon, changing climate, climate change, climate variation, environmental change, global warming, global climate, greenhouse effect, greenhouse gas, nuclear winter, ocean acidification, refrigeration effect, sea level rise, and warming climate. The search terms were chosen to account for various phrases used by experts over time, uncertainty in future scenarios among earlier textbooks, and the use of alternative phrases to avoid social or political controversy.

To limit the selection to anthropogenic climate change, we excluded passages that primarily focused on the carbon cycle, atmospheric composition, ozone depletion, photosynthesis, fluctuation of seasonal temperatures, or climate change events in past geologic eras. In most

textbooks, headings marked the start and end of the passage; however, in earlier textbooks, the passage transitioned without a delineating header. Occasionally, a climate change passage incorporated non-relevant topics (e.g., the carbon cycle or photosynthesis). Since these sentences were within the delineated passage, we included them in our counts, which inflated the length of the passage. Rarely were anthropogenic climate change and the greenhouse effect addressed in multiple locations within the book. In these instances, we averaged the two start pages and coded the selections separately, though data were combined for processing. This situation inflated the length of the passage since the descriptions of climate change were repeated in each section.

Text outside the main passage (e.g., a side panel) was included if it contained at least three paragraphs. We counted shorter passages only if there was no other content, as was often the case for textbooks published in the 1970s. Climate change content was often included in sidebars or supplemental page panels in 1980s textbooks.

To calculate the start location as a percentage of the books' total pages, we used the last page of the last chapter since appendices, indices, and glossaries varied substantially between books and over time. In the case that a textbook contained no passage related to climate change, we excluded that book from the start-page average.

## Text analysis

Sentences within the climate change passage were counted and categorized. We included only the main body of the text; we did not include figure captions, review questions, lists of learning objectives, chapter summaries, scientist biographies, or footnotes. We also did not count rhetorical questions (i.e., "How does this happen?" or "Why?") since these inflated the number of sentences without addressing content. Sentences within parentheses were not included, as they usually referred to other content or chapters.

Each sentence was classified into one of three broad categories: 1) a general description of the greenhouse effect, 2) impacts of climate change, or 3) actions that ameliorate the effects of climate change. Sentences addressing the impacts of climate change were further delineated into nine subcategories: 2a) sea level, glaciers, or polar ice; 2b) rainfall or drought; 2c) extreme weather events (e.g., hurricanes, wildfires, el niño); 2d) temperature changes; 2e) species extinction or range shifts; 2f) agricultural impacts; 2g) ocean acidity or currents; 2h) human health; and 2i) "other." Some sentences addressed two or three subcategories; in this case we placed a portion of one point in each of the addressed subcategories. If a single sentence included more than three subcategories, we tagged it as "other." Also, action sentences were further divided into 3a) individual/local actions or 3b) national/international actions. Rarely did a text include generalized or vague sentences addressing solutions; we categorized these as "other." Individual actions were further categorized as high-impact (e.g., reproductive planning, transportation, diet choices) or low-impact (e.g., recycling, turning off lights) for reducing carbon emissions, according to Wynes and Nicholas [23].

## Figure analysis

Any data figures, tables, charts, or graphs referenced in the text passage were included in the analysis (hereafter referred to as figures). Data categories included global temperatures, carbon dioxide levels, sea level change, species movement (often maps), and photographic evidence (e.g., comparison in size of glaciers over time), or "other" (e.g., agricultural yields, changes in rainfall). If a single figure depicted multiple data categories, we included a portion of one point in each of the addressed categories. For example, many textbooks included a graph showing the rise in $CO^2$ levels and global temperatures over time. In this case, each category received a

count of one-half point. We did not include descriptive figures, such as a photograph of an impacted animal or an infographic showing how greenhouse gases block infrared energy from leaving the atmosphere, in the data figure count.

### Intercoder reliability

Each author coded each textbook separately. Agreement for each data entry was defined as the same score for numbers under 10, or scores within ±1 for counts of 10 and above. We used a simple percent agreement due to the large number of possible categories. Agreements of 90% or higher were considered acceptable [24]. For any book with an intercoder agreement of less than 90%, the authors recoded the passage together. In almost every instance, 100% agreement was attained though discussion and consensus; for the few data points without agreement, we averaged the two scores.

## Results

Our data indicate that climate change coverage changed substantially over time, the greatest shift occurring in the 1990s. Modifications to content since 2000 have been minor and emphasize the impacts of climate change while deemphasizing solutions.

### Amount of content

Coverage of climate change in textbooks increased from 1970 to 2019, though not consistently or relative to scientific research or the severity of the issue. Prior to 1990, the textbook passage addressing climate change averaged fewer than 11 sentences. In the 1990s, coverage expanded to almost 40 sentences, a 264% increase. However, two outliers with large passages, at 90 and 108 sentences, were two editions of the same book. The median content during this decade was 30 sentences. The following decades increased the average climate change content from 51 sentences in the 2000s to 67 sentences in the 2010s. The decade spanning 2010 to 2019 also showed a large range, with books presenting between 22 and 107 sentences. The median length of the passage dropped from 52 sentences in the 2000s to 45 sentences in the 2010s (Fig 1).

The coverage of climate change in textbooks has decreased compared to the number of scientific articles published after the 1990s. In the 1990s, there was approximately one textbook sentence for every 200 scientific publications about climate change. In the 2010s, the ratio was one textbook sentence for every 1,100 scientific publications (Fig 2A). Textbook coverage has remained proportional to global temperatures. In the 1990s, there was approximately one textbook sentence for each 0.01˚ Celsius increase in global temperature. Since 2000, the textbook content decreased slightly in comparison to global temperatures, although the median number of sentences in the 2010s dropped to almost half the temperature anomaly (Fig 2B). The years 2015–2019 show a notable increase in global temperatures. The average number of textbook sentences would need to increase by 23 sentences to remain in proportion to these temperatures.

### Start location of climate change passage

Climate change coverage has been consistently presented at the end of biology textbooks. Small shifts in placement over time have moved the topic further toward the end of the book. The earliest start location, at a median of 85.3% of the book, occurred in the 1970s. The latest start location, at a median of 97.6%, appeared in textbooks published in the 2010s (Fig 3). Only four textbooks, published between 2002 and 2014, contained multiple passages addressing climate change. We averaged the start locations, which remained toward the end of the book at 85.6%.

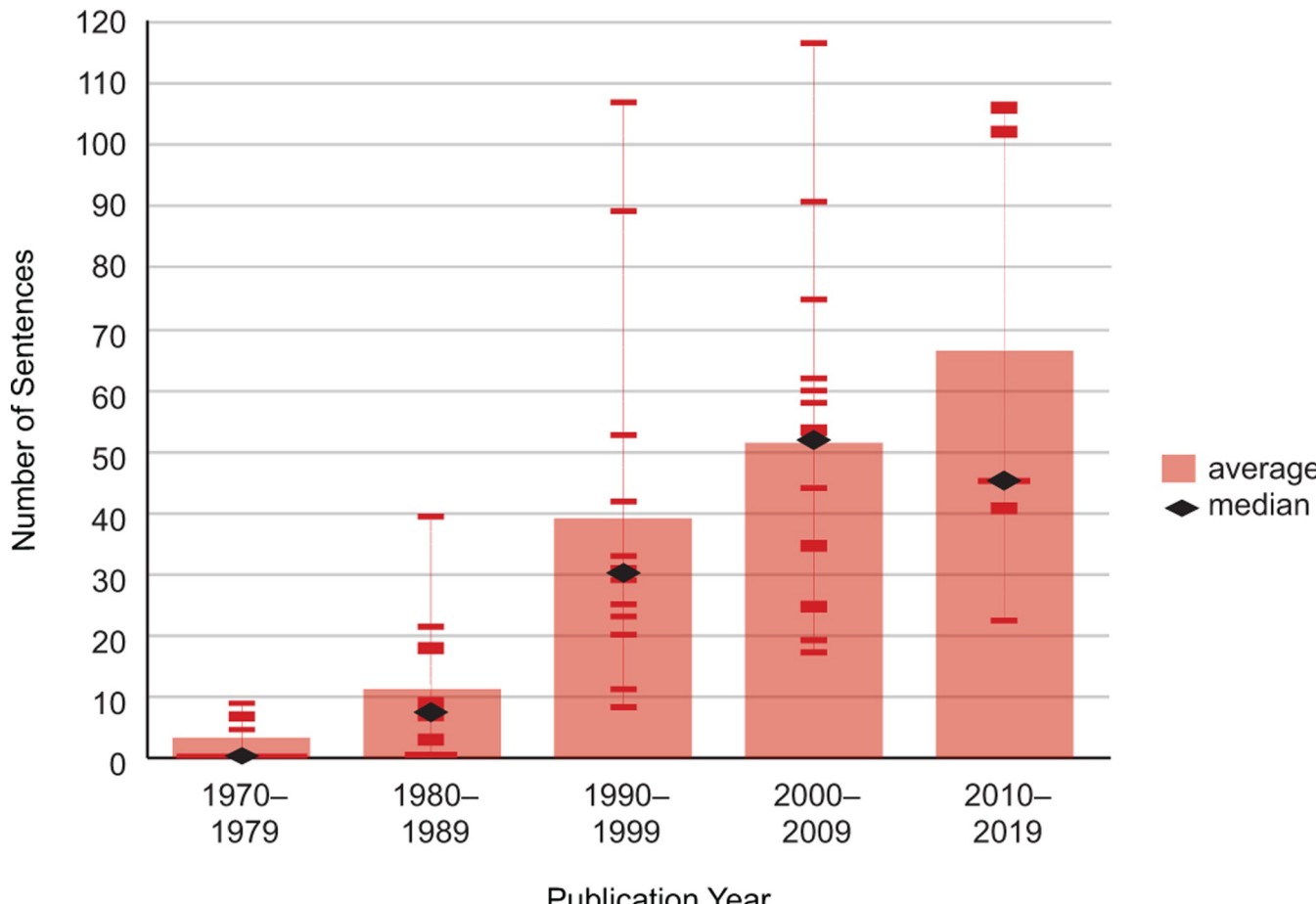

**Fig 1. Amount of coverage addressing climate change in college introductory biology textbooks.** The average amount of coverage increased over time, though the largest increase occurred in the 1990s. Due to the large range in passage length, the median number of sentences for each decade was calculated. The 1990s still showed the largest increase in passage size, though textbooks published in the 2010s decreased in median length.

## Sentence topic analysis

Each sentence in the passage about climate change was coded as: 1) a description of the greenhouse effect, 2) effects of climate change, or 3) actionable solutions for the problem. In the 1970s textbooks, the content was almost exclusively a description of the greenhouse effect (Fig 4). The description sentences decreased to approximately 50% of the content in the 1990s, remained stable through 2009, and then decreased to less than 40% in the last decade. The average number of description sentences was the same in 2000–2009 and 2010–2019 (23.6 and 23.7, respectively), yet the passage grew overall due to the number of sentences addressing effects of climate change (an average of 22.4 in the 2000s and 41.4 in the 2010s). Coverage of the consequences of climate change in the most recent books is almost double the description of the greenhouse effect. Sentences offering actionable solutions peaked in the 1990s, at 15.5% of the passage, and declined in the most recent books, to 3.1% of the passage.

We further categorized the effects of climate change addressed in textbooks published from 1990 to 2019 (earlier decades had few sentences addressing the consequences of climate change). Impacts on extinction rates made up the largest number of sentences in all three decades, and the topic increased over time. The coverage of climate's impact on sea level rise, agriculture, rainfall, and temperature all decreased over time, while coverage of human health

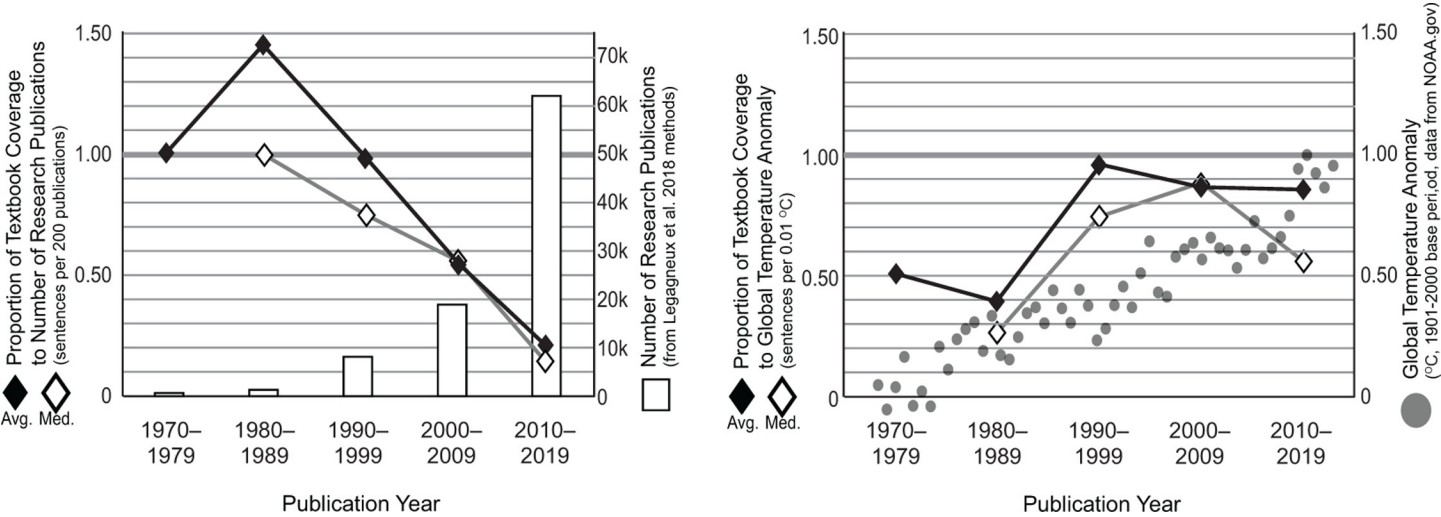

**Fig 2.** Comparison Between Textbook Coverage of Climate Change and (a) Research Publications, (b) Global Temperatures. (a) The number of research publications between 1970 and 2019 has grown exponentially, but the proportion of textbook coverage has decreased from one textbook sentence per 200 scientific publications to one textbook sentence per 1,100 scientific publications. (b) Textbook coverage and global temperature anomalies have grown more proportional over time, though the 2010–2019 median textbook coverage declined sharply compared to rising global temperatures.

concerns increased substantially. Sentences addressing extreme weather events remained stable, and ocean topics remained among the least covered, though the coverage is increasing (Fig 5).

We also further analyzed sentences related to actionable solutions for climate change. Consistently, actions that focused on national or international responsibilities were presented over four times more than individual or local solutions.

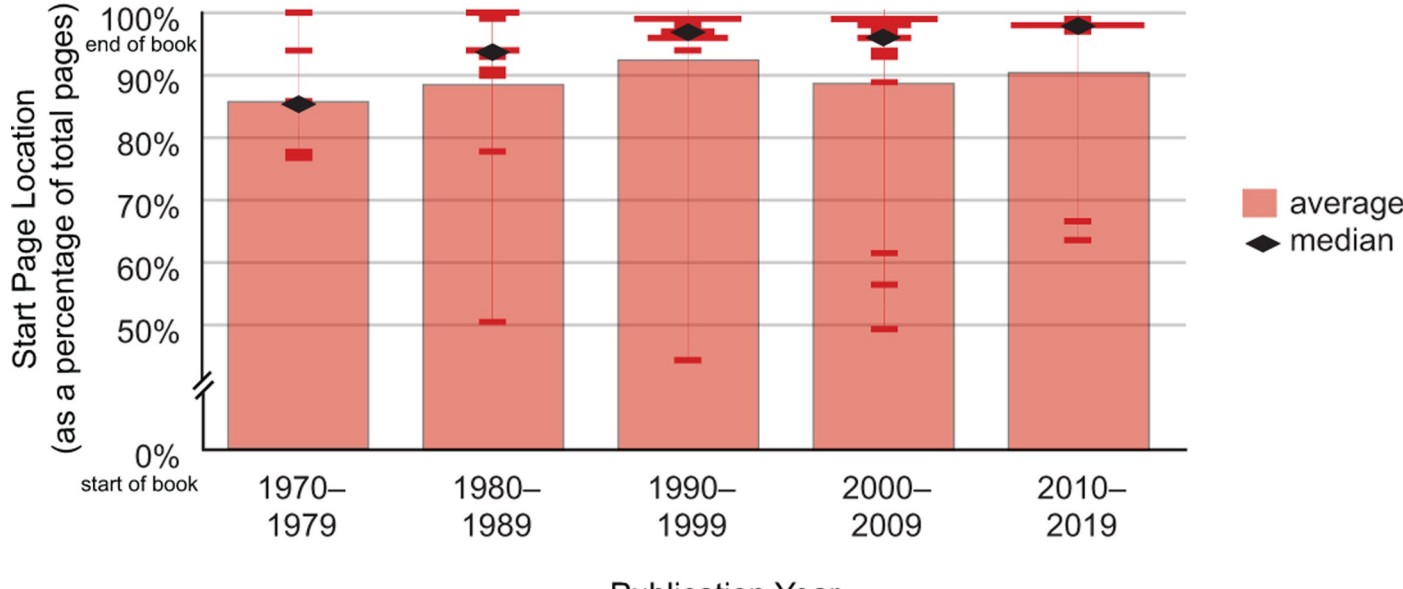

**Fig 3. Location of passage addressing climate change in college introductory biology textbooks.** Climate change coverage has been consistently presented toward the end of the textbook, though has moved further toward the end in more recent books. The median start location, ranged in the last 2–5% of the book over the last three decades.

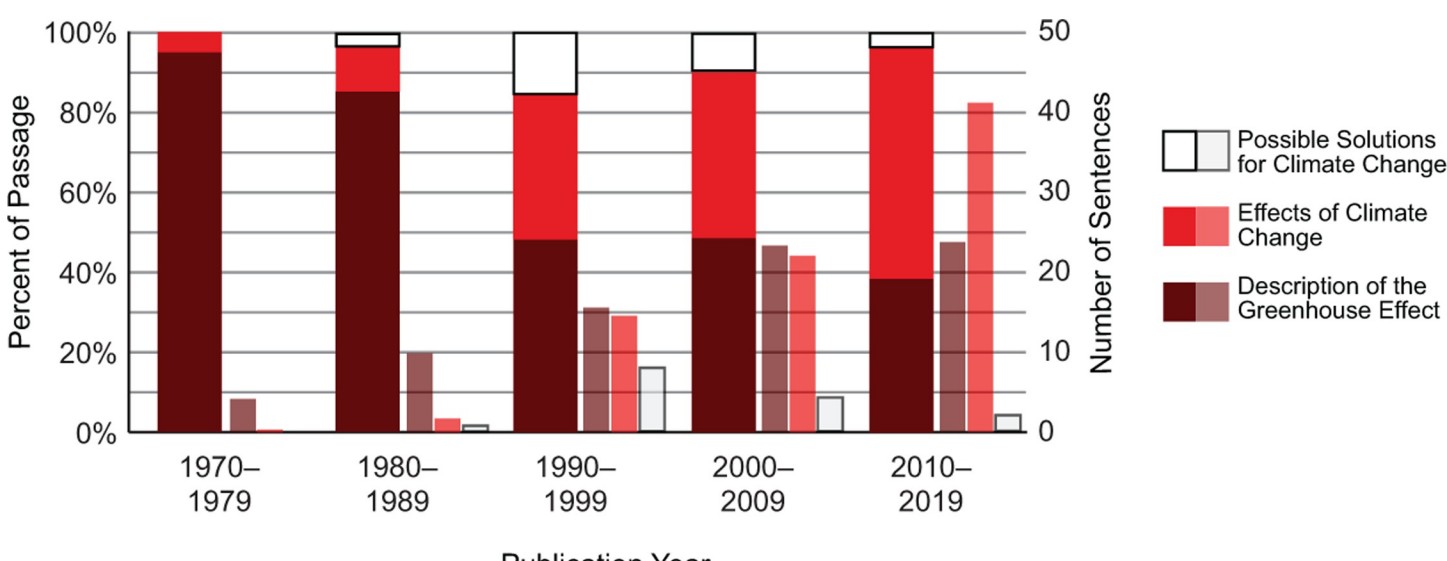

**Fig 4. Sentences dedicated to description, effects, and solutions within the climate change passage of college introductory biology textbooks.** Coverage of the effects of climate change increased substantially over time so that, in the most recent books, it outweighs the description sentences and is responsible for the growth of the passage. Sentences offering actionable solutions peaked in the 1990s, at 15% of the passage, and declined to 3% of the passage in 2010–2019 books.

Within individual- or local-level solutions for climate change, most actions described less influential behaviors, such as recycling. Only eight books addressed transportation as a way of decreasing the release of greenhouse gasses; no textbook addressed dietary choices, housing choices, or family planning (Fig 6).

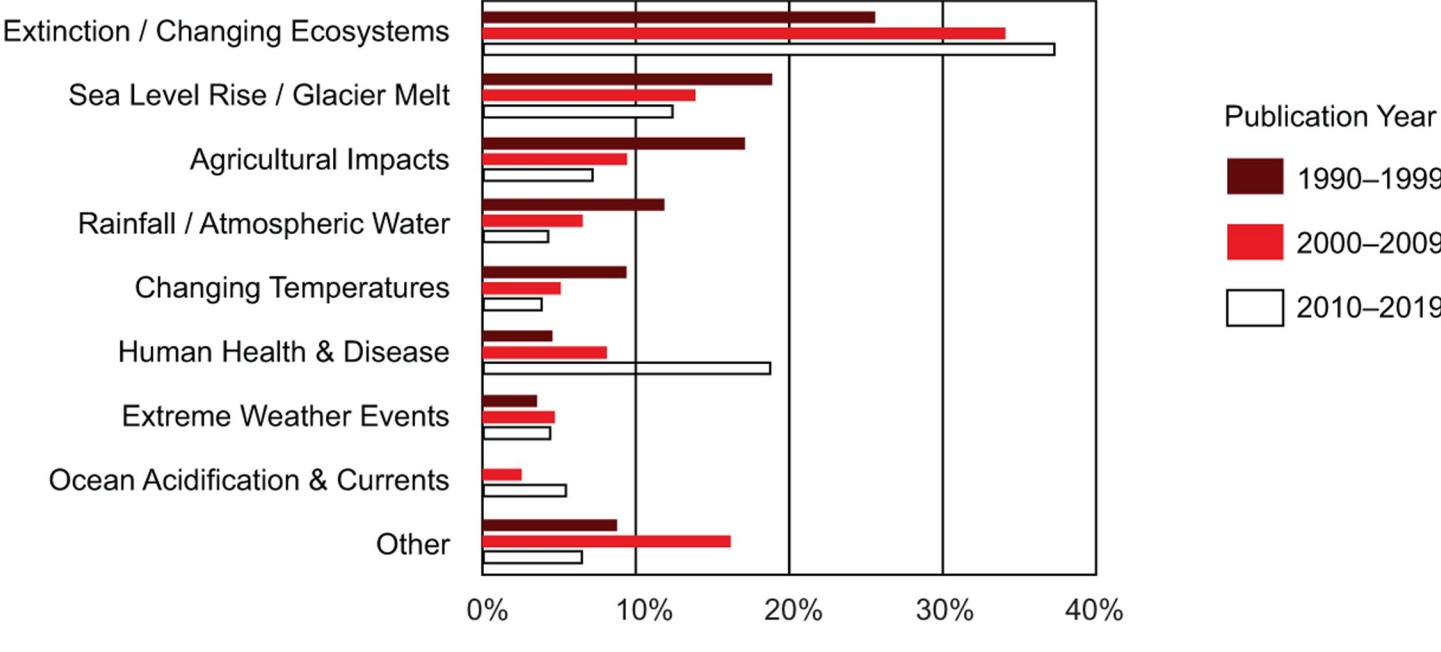

**Fig 5. Specific effects of climate change addressed within the climate change passage of college introductory biology textbooks.** Sentences focused on the effects of climate change most often addressed ecosystem changes or extinction. Human health sentences have increased every decade. Other categories of climate change effects, such as sea level rise, agricultural impacts, rainfall, and changing temperature decreased every decade.

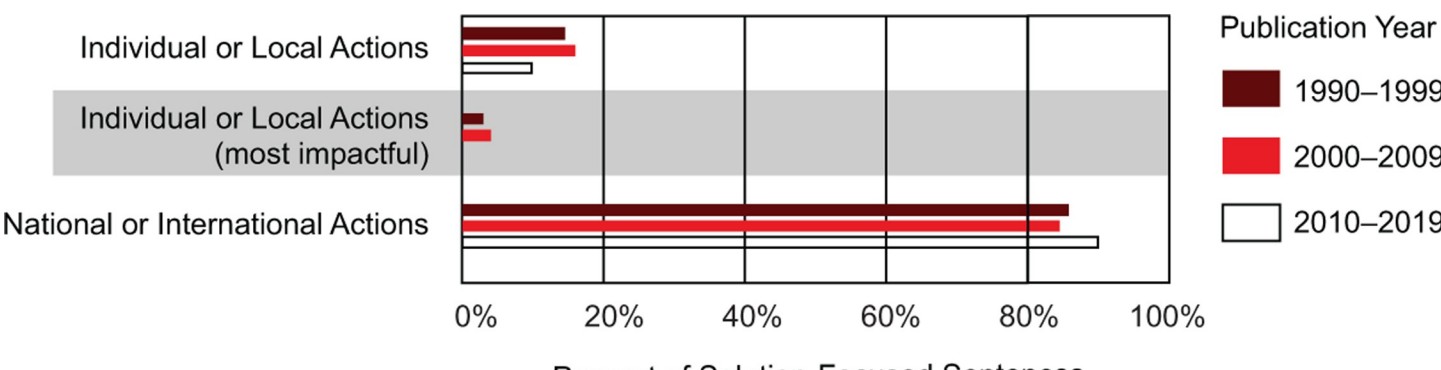

**Fig 6. Specific solutions for climate change addressed within the climate change passage of college introductory biology textbooks.** Textbooks have focused on national and international actions, averaging over 80% of the solution-focused sentences. Sentences addressing individual or local actions were few. Of those, most suggestions described behaviors with a low impact on carbon emissions.

### Figure analysis

While textbook passages about climate change appeared in the 1970s, few data visuals were presented until the 1990s. The number of data figures increased over time but have remained small (from an average of 1.2 data figures in the 1990s to an average of 2.6 data figures in the 2010s).

The most common data figures are those that display $CO^2$ levels and global temperatures. Almost every textbook published after 1988 showed a combined figure of these data series. These figures peaked in the 1990s and 2000s. In the 2010s, qualitative photographic evidence (e.g., glacial melt) sharply increased. Maps showing the impact of climate change on species movement also appeared in the 2000s and increased through the 2010s (Fig 7).

## Discussion and conclusion

After analyzing the climate change content from five decades of introductory college biology textbooks, our findings indicate that the amount of content, placement within the book, and communication of solutions have not kept pace with the severity or scope of the problem. However, coverage of the effects of climate change, in text passages and figures, has increased and diversified over time.

### Amount of content

We expected the amount of content to increase in proportion with publications on climate change and the severity of societal and environmental consequences. Despite the large growth in climate change research, funding, and journalism since the early to mid-2000s, textbooks have not reflected the same increased emphasis. The average size of the passage on climate change grew more slowly in the 2000s; the median length of the passage surprisingly decreased in the 2010s. The stagnancy in climate change coverage may have multiple influences, including societal concerns, student mental wellness, and author interest.

First, societal influence could have played a role in the decrease of content. The Kyoto Protocol, Earth Day 2000, the film *An Inconvenient Truth*, multiple UN climate change conferences, and IPCC publications increased total media coverage of climate change from 1998 through 2010. This attention, however, was often met with backlash or controversy (e.g., 2010 IPCC email scandal, NSTA decline of 50,000 free copies of *An Inconvenient Truth*, or the political conservative alignment with the fossil fuel industry) [25]. Given the role of Texas in fossil

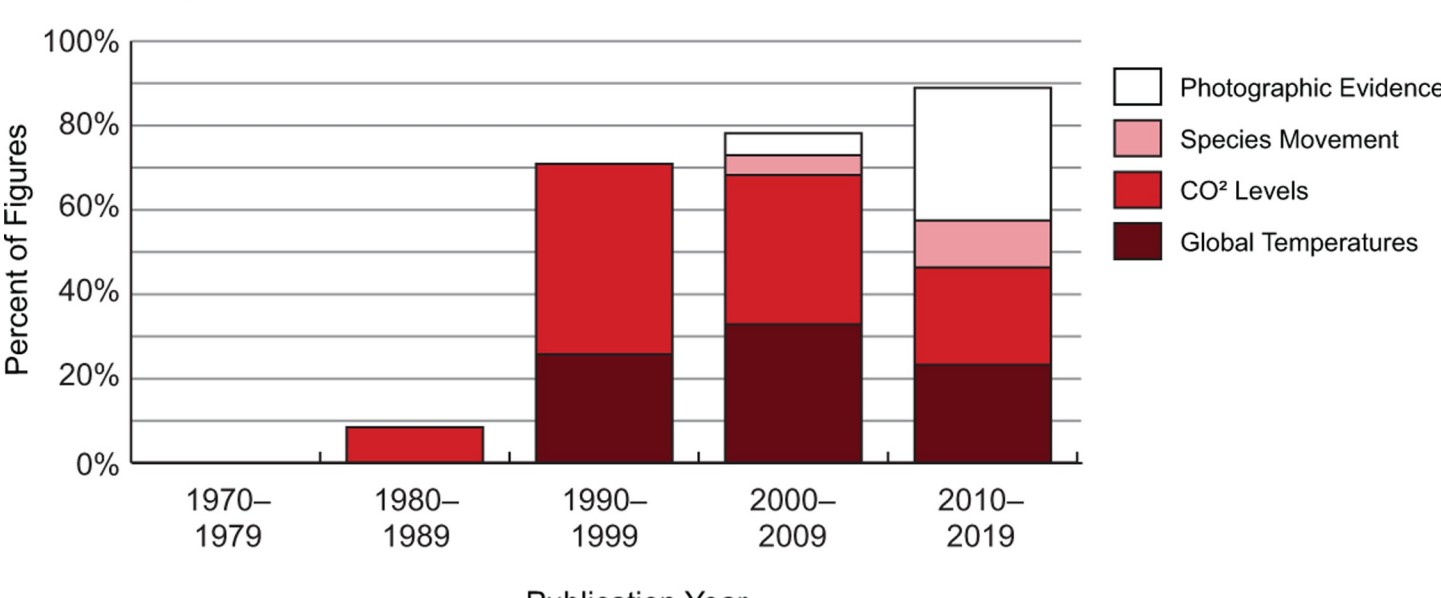

**Fig 7. Types of figures presenting data in textbook passages about climate change.** Early textbooks contained no data visuals. Most textbooks published in the 1990s included a graph showing carbon dioxide levels and Earth temperatures until 2010. Recent textbooks include more diversity in data figures, such as photographs of glacier retreat and maps of species movements.

fuel production and its importance to textbook publishers (known as The Texas Effect), minimizing content on climate change could reduce problems in textbook adoption [26]. Publishers often diminish controversial issues to tailor textbook content, reflecting public concerns [27]. Due to the history of biology textbooks and their inclusion of controversial topics toward the rear of the books, societal concerns may have contributed to the stagnancy of climate change coverage.

Second, the amount of content may have decreased as authors or publishers grew concerned about the negative psychological impacts of a focus on climate change. Extreme weather events, uncertainty, and guilt can trigger anxiety, depression, grief, trauma, and suicidal ideation [28]. However, our analysis of the topics addressed within the climate change passage shows an emphasis on the effects and a decline in solutions, which would increase feelings of hopelessness and helplessness in students. Therefore, we refute the contribution of "climate despair" on the stagnancy in climate change content.

Finally, textbook authors may weigh topics more heavily if the subjects reflect their research interests or expertise. There has been a notable and dramatic decrease in the number of organismal- and ecological-based college classes required for biologists, as well as a decline in natural history careers, grants, and research among faculty [29, 30]. This would result in fewer faculty with expertise in ecology or organismal biology, and more authors of recent biology textbooks focused on cellular and molecular biology. To check the legitimacy of this claim, we searched for the research focus or dissertation topic of the first two authors from each textbook published in the 1990s, at the height of climate change coverage, and in the 2010s, with the drop in climate change coverage. In the 1990s, nine authors (53% of those with scientific foci) studied organismal biology or ecology, five studied physiology, and three (18%) studied cell or molecular biology. There were an additional eight authors focused on science education or science writing. In the 2010s, five authors (45%) studied organismal biology or ecology, one studied

physiology, and five (45%) studied cell or molecular biology. Only two other authors focused on science education or science writing. This increase in the percentage of authors who studied cell or molecular biology, or the decline in authors focused on science communication, supports the idea that the author's expertise and interest may play a role in the textbook's coverage of environmental issues.

## Location of climate change passage in textbooks

We predicted that the start location of the passage addressing climate change would move forward in the book, as an indication of its growing importance. However, the start location of the passage moved toward the rear of the book.

Biology textbooks usually organize content from small-scale concepts (i.e., molecular biology and cell anatomy) to large-scale topics (i.e., evolution and ecology). However, this organization dates from the 1930s when biology, rather than organismal topics of botany and zoology, took hold in the curriculum. The 1934 textbook *Biology for Today* by Curtis, Caldwell, and Sherman was the first to group conservation, reproduction, and evolution at the back of the book [16]. These topics are arguably most controversial and, given the documented omissions of end chapters and the timing of this arrangement (the 1930s marked a turning point in biology textbooks due to the anti-evolution movement, eugenics and other race-related topics), it is likely that the placement allowed these topics to be easily excluded. Over time, reproduction shifted forward in the book to be addressed with heredity or physiology, and evolution moved forward to be addressed following cellular topics. These shifts left conservation issues as the last topic covered in the book.

## Climate change content analysis and figure analysis

We found mixed outcomes in our analysis of content. Passages shifted from providing only a simple description of the greenhouse effect to including more coverage of the effects of a warming climate. Sentences incorporated a greater range of effects, mirroring the data presented in the passage's figures. The increased variety of effects of climate change, presented through examples and data figures, could improve students' acceptance of climate change through inductive reasoning.

However, content addressing actionable, productive solutions peaked in the 1990s and has decreased in the past two decades. This creates an uncomfortable message for students, that climate problems are large and without solutions, and reinforces helplessness, anxiety, and depression [31].

Additionally, the solutions to climate change addressed in textbooks focus on national or international actions, leading students to believe that their individual actions are inconsequential. While a single person has a small impact on carbon emissions, textbooks and other instructional materials contribute to the education of millions of young people. Awareness of industrial carbon contributions could change consumer practices and impact corporate decision-making. Concentrating on political solutions is disconnected from the textbook's audience since fewer than half of 18-25-year-olds engage in political elections [32, 33]. However, family planning, transportation and housing options, and food choices are prominent issues for this age group and are among the most influential individual behaviors on carbon emissions.

While we acknowledge that textbooks emphasize descriptions of concepts and processes rather than advice on personal behaviors, there are examples of such content dealing with other biological topics. Textbooks of the early 20[th] century frequently included advice about hygiene, agriculture, and nutrition. Modern textbooks often include sidebars or breakout boxes on topics such as sexually transmitted disease, skin cancer, endocrine disrupting

chemicals in food and household products, or genetic testing, all of which include aspects of behavioral choice. Therefore, there is precedent for students to be presented with a selection of individual behaviors with high impact on climate change.

## International impacts of U.S. biology textbooks

This study focused on U.S. textbooks. However, the globalization of the publishing industry since the 1990s has reduced and conglomerated textbook publishing companies to the "big three:" McGraw-Hill, Pearson, and Houghton Mifflin Harcourt [34]. McGraw-Hill (North America) and Pearson Education (North America & International divisions) dominate the science textbook market. Both offer products to most countries and in scores of languages.

Biology, as compared subjects such as history, culture, or language, is not limited to a nation's borders or societal constructs. Therefore, translated biology textbooks can be distributed among international audiences with few changes. Most international editions of textbooks are identical in content and pagination to the U.S. versions. Therefore, we propose that our findings may be generalizable outside the U.S. textbook market.

## Further recommendations

Education is the single strongest predictor of public awareness of climate change [35]. Climate change is also a tremendous threat to human society. Given Americans' outsized contribution to the problem and resistance to accept anthropogenic climate change, we propose that U.S. educators or curriculum developers consider the implicit messages sent through the amount of coverage, inclusion of subtopics, and placement of climate change information in instructional materials or course organization.

While textbook use has decreased in recent years, textbooks can serve as documents to analyze the material presented in introductory biology courses. We recognize that student textbook usage continues to decline, yet we consider the following suggestions should apply to any instructional methods.

We also acknowledge that this study is limited in content level and subject area. The topic of climate change branches beyond the sphere of science into politics, sociology, and economics. In our focus on biology and college-level formal education, we may have missed other domains where young people are exposed to important lessons in climate change.

We offer four recommendations to address the concerns about climate change coverage raised through this research. These recommendations also provide opportunities for further research, as any changes could be applied using experimental methods.

First, we suggest that authors, publishers, and educators reconsider the standard order of topics within biology courses. Rather than starting with subcellular topics and expanding out to organismal and, finally, ecological topics, educators could start with large-scale topics and drill down to cellular processes. Alternatively, instructors could apply scaffolding to begin with organism-level content, the level at which students interact with the natural world, and then expand to ecosystems; after revisiting the organism-level, the class could then dive into cellular and subcellular topics. Whether the standard organization is in place due to tradition or concern about controversy, the importance of environmental issues may require a reevaluation of the order of course topics.

Second, we propose that, rather than providing extensive coverage of climate change effects with few actionable solutions, authors and educators intentionally pair effects with solutions. This allows students to focus on positive engagement with the topic of climate change, while accepting and adapting to global warming. The most extensive coverage of climate change in 2010–2019 included just 107 sentences. This content was found on 4 pages in a 1,279-page

textbook. The severe consequences of climate change on human society and ecological stability warrants a substantial increase in coverage.

Third, we encourage publishers or developers of instructional materials to recruit ecologists, environmental researchers, or science communicators as authors for educational materials. While there are many important scientific topics to address in introductory biology, developers of instructional materials ought to ensure a balance between organismal, ecological, cellular and molecular biology. For example, the AP Board lists eight units addressed in its AP Biology course framework (used as a substitute for introductory college biology). The first six of the eight units deal with subcellular concepts. The final unit is ecology. This conveys a message that ecology is an afterthought with substantially less information to learn. Quantitatively analyzing the topics of instructional materials could result in more balanced instructional materials.

Lastly, we recognize that many of these issues are woven into the culture of the biological science community. We hope this paper adds to the growing evidence that ecological problems created by humanity are, at least partially, the result of this culture. We recommend college biology faculty, administrators, and granting agencies reflect on departmental courses, faculty research areas, and awards. Natural history and organismal biology have declined as areas of faculty expertise and required course offerings despite serving as excellent recruitment courses for students. While much of this shift occurred due to exciting discoveries in cellular and subcellular biology, the recent developments and severe consequences of environmental issues warrant increased awareness and appreciation of organisms and ecosystems throughout our education system.

## Supporting information

**S1 Appendix.**
(DOCX)

## Acknowledgments

This project was aided by the Provost's Professional Experience Program at North Carolina State University. Special thanks to Shaun Bennet and the Data Projects Unit from North Carolina State University, Kendall Ross from California State University, and Brice Sullivan from the University of California in Los Angeles.

## Author Contributions

**Conceptualization:** Jennifer M. Landin.

**Data curation:** Rabiya Arif Ansari, Jennifer M. Landin.

**Formal analysis:** Rabiya Arif Ansari, Jennifer M. Landin.

**Investigation:** Rabiya Arif Ansari, Jennifer M. Landin.

**Methodology:** Rabiya Arif Ansari, Jennifer M. Landin.

**Project administration:** Jennifer M. Landin.

**Resources:** Rabiya Arif Ansari, Jennifer M. Landin.

**Supervision:** Jennifer M. Landin.

**Visualization:** Jennifer M. Landin.

**Writing – original draft:** Rabiya Arif Ansari, Jennifer M. Landin.

**Writing – review & editing:** Jennifer M. Landin.

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
