## [Decision Letter · Decision Letter 0]

9 Sep 2022

PONE-D-22-22659Coverage of Climate Change in Introductory Biology Textbooks, 1970-2019PLOS ONE

Dear Dr. Landin,

Thank you for submitting your manuscript to PLOS ONE. After careful consideration, we feel that it has merit but does not fully meet PLOS ONE’s publication criteria as it currently stands. Therefore, we invite you to submit a revised version of the manuscript that addresses the points raised during the review process.

In particular, please pay close attention to Reviewer 1's comments on clarifying the focus of the paper at the introduction and establishing a clearer connection between the extant literature and your predictions in these textbooks, and Reviewer 2's comments on better conceptualizing your argument and (at least) outlining implications for non-US contexts. 

We look forward to receiving your revised manuscript.

Kind regards,

Charles (Charlie) Jonathan Gomez

Academic Editor

PLOS ONE

3. We noted in your submission details that a portion of your manuscript may have been presented or published elsewhere. One graph from this work (Fig 7) has been requested for a book, Storytelling to Accelerate Climate Solutions, scheduled for publication by Springer in 2022-2023.Please clarify whether this publication was peer-reviewed and formally published. If this work was previously peer-reviewed and published, in the cover letter please provide the reason that this work does not constitute dual publication and should be included in the current manuscript.

Reviewers' comments:

Reviewer's Responses to Questions

**Comments to the Author**

1. Is the manuscript technically sound, and do the data support the conclusions?

Reviewer #1: Yes

Reviewer #2: Yes

2. Has the statistical analysis been performed appropriately and rigorously? 

Reviewer #1: Yes

Reviewer #2: Yes

3. Have the authors made all data underlying the findings in their manuscript fully available?

Reviewer #1: No

Reviewer #2: Yes

4. Is the manuscript presented in an intelligible fashion and written in standard English?

Reviewer #1: Yes

Reviewer #2: Yes

5. Review Comments to the Author

Reviewer #1: Thank you for the opportunity to review this paper. The study presents interesting and highly valuable results on the coverage of climate change in college biology textbooks between 1970 and 2019. The sampling of textbooks and their analysis are described with great detail. The paper is written well, and it is a pleasure to read.

I have some suggestions for minor revisions. First, the introduction in its current form provides a lot of interesting information about climate change, but it doesn’t explain what the paper will focus on and what will be studied. While this is described in the abstract, it would be helpful to guide the reader also in the introduction. Now the focus of the study is revealed only at the end of page 5.

Second, the section on science textbooks as historical documents is an interesting read and motivates the study nicely. Yet a clearer connection between the cited literature and what the authors predict they will find in the textbooks would be helpful. This could be one or two sentences clearly stating the connection between the literature and the authors’ predictions.

Third, while the data was made available, the link didn't work for me. It would be good to check that it's correct and works.

Finally, I would like to note that I really appreciated the discussion on practical recommendations at the end of the paper. Although the use of textbooks may be reducing, textbooks still provide interesting material for analyzing education’s role in sharing knowledge and developing understandings among the future generations concerning societal challenges such as the climate change.

Reviewer #2: The article is based on a review of college level introductory biology textbooks, with a focus on how the focus on climate change has changed in the period 1970-2019. The main finding of the study is that, and quite contrary to the outlined hypothesis of the study, is that the focus on climate change, has declined over the years. In addition, focus seems to have changed to climate change as a problem rather than on discussing solutions to climate change, especially at local and national levels. The article has a clear structure and language, and is seemingly based on thorough statistical (descriptive) analyses. If the following shortcomings and unclarities are addressed, it would clearly add new knowledge to the research field of climate change education:

Conceptually the article is less elaborated upon. As the argumentation moves forward it becomes clear that contextual factors are likely key explanatory factors for why climate change in textbooks (and education) change (e.g., line 82-86; 334-344), e.g. politics, social and cultural. Also, educational policies could be relevant to include here. Maybe curriculum theory could help the authors in including this important factor in the analysis (see e.g. Goodlad 1979: 'Curriculum inquiry: The study of curriculum practice')? This demonstrates well the interlinkages between policy planning (ideological level), elaboration (the actual), operational (teaching materials, such as textbooks), perceived (how perceived by end users, the teachers), and experienced (by the students). This could potentially also provide a meaningful explanatory model for how the operationalized textbooks are linked to trends in the society as a whole.

This feeds into the second major reservation that I have: How valid/relevant are the findings to contexts other than the US? A positioning of the paper into a wider scientific context would be very relevant and recommended, and reservations with regards to the generalizability of the data warranted.

Some of the arguments and conclusions also seem a bit shallow and one-dimensional. For example, there is not necessarily a one-way relationship between quantitative and qualitative contents of textbooks, i.e., the number of times words/concepts, etc. are used/mentioned in the books. The question/issue should also be seen within a wider educational context of other subjects, such as social studies, etc. that the students are exposed to through their educational course.

6. PLOS authors have the option to publish the peer review history of their article (what does this mean?). If published, this will include your full peer review and any attached files.

Reviewer #1: No

Reviewer #2: No

---

## [Author Response · Author response to Decision Letter 0]

23 Oct 2022

Dear Dr. Gomez and esteemed reviewers,

Thank you for the time you spent reviewing our paper, and for the thoughtful and helpful comments on our manuscript. We hope that our revisions reflect our appreciation for your expert review.

Reviewer 1

We greatly appreciate the suggestion to introduce the topic of study earlier in the introduction. We have included a paragraph at the start of the introduction (lines 39-43) to clarify the focus and address the organization of that section.

We have refined each of the four hypotheses to clarify connections between the “textbooks as historical documents” section and our predictions (lines 123-146). 

Also, we have corrected a setting to make the dataset publicly available. Please accept our apologies for the error and our thanks for pointing out this problem. You should be able access the data at: https://doi.org/10.5061/dryad.tht76hf2f

We humbly offer our thanks for such positive and constructive notes about our paper.

Reviewer 2

Thank you for the suggestions to elaborate on our argument and broaden our conclusions for an international audience. We have made the following changes and believe they have greatly improved the paper.

To improve the conceptualization of our argument, we added two paragraphs attempting to address curricular theory and decision-making and its relevance to our argument prior to our predictions (lines 112-120). Thank you for guiding us to the Goodlad book and associated curriculum theories.

We wholeheartedly agree with the premise that generalizability is warranted. We added two paragraphs under an additional header (“International Impacts”) in the conclusions section (lines 441-451). 

Lastly, we agree that the topic of climate change is more broadly addressed in the educational system. As such, a full analysis of the topic would need to be much deeper and more complex than we have presented. However, we were concerned about addressing a wider educational context involving other subjects since our sample of textbooks was limited to introductory biology. Therefore, we have added an acknowledgement of the limited nature of our study (lines 463-466). We hope that this acknowledgement is sufficient to address your justified concern. Thank you for your assistance and guidance in developing a more nuanced paper.

---

## [Decision Letter · Decision Letter 1]

18 Nov 2022

Coverage of Climate Change in Introductory Biology Textbooks, 1970-2019

PONE-D-22-22659R1

Dear Dr. Landin,

We’re pleased to inform you that your manuscript has been judged scientifically suitable for publication and will be formally accepted for publication once it meets all outstanding technical requirements.

Kind regards,

Charles (Charlie) Jonathan Gomez

Academic Editor

PLOS ONE

Additional Editor Comments (optional):

Reviewers' comments:

Reviewer's Responses to Questions

**Comments to the Author**

1. If the authors have adequately addressed your comments raised in a previous round of review and you feel that this manuscript is now acceptable for publication, you may indicate that here to bypass the “Comments to the Author” section, enter your conflict of interest statement in the “Confidential to Editor” section, and submit your "Accept" recommendation.

Reviewer #1: All comments have been addressed

Reviewer #2: All comments have been addressed

2. Is the manuscript technically sound, and do the data support the conclusions?

Reviewer #1: Yes

Reviewer #2: Yes

3. Has the statistical analysis been performed appropriately and rigorously? 

Reviewer #1: Yes

Reviewer #2: Yes

4. Have the authors made all data underlying the findings in their manuscript fully available?

Reviewer #1: Yes

Reviewer #2: Yes

5. Is the manuscript presented in an intelligible fashion and written in standard English?

Reviewer #1: Yes

Reviewer #2: Yes

6. Review Comments to the Author

Reviewer #1: The authors have adequately responded to my feedback from the previous round of review. The response letter to the reviewers as well as the revised manuscript explain the recommended changes. As all identified concerns have been addressed, I can recommend the manuscript for publication.

Reviewer #2: All comments seem to have been taken into account, and addressed in appropriate and well argued ways.

7. PLOS authors have the option to publish the peer review history of their article (what does this mean?). If published, this will include your full peer review and any attached files.

Reviewer #1: No

Reviewer #2: No

---

## [Editor Report · Acceptance letter]

24 Nov 2022

PONE-D-22-22659R1 

Coverage of climate change in introductory biology textbooks, 1970-2019 

Dear Dr. Landin:

I'm pleased to inform you that your manuscript has been deemed suitable for publication in PLOS ONE. Congratulations! Your manuscript is now with our production department. 

Kind regards, 

on behalf of

Dr. Charles (Charlie) Jonathan Gomez 

Academic Editor

PLOS ONE